# 3D printing of robotic soft actuators with programmable bioinspired architectures

Manuel Schaffner [1], Jakob A. Faber [1], Lucas Pianegonda[1], Patrick A. Rühs [1], Fergal Coulter [1,2] & André R. Studart [1]

Soft actuation allows robots to interact safely with humans, other machines, and their surroundings. Full exploitation of the potential of soft actuators has, however, been hindered by the lack of simple manufacturing routes to generate multimaterial parts with intricate shapes and architectures. Here, we report a 3D printing platform for the seamless digital fabrication of pneumatic silicone actuators exhibiting programmable bioinspired architectures and motions. The actuators comprise an elastomeric body whose surface is decorated with reinforcing stripes at a well-defined lead angle. Similar to the fibrous architectures found in muscular hydrostats, the lead angle can be altered to achieve elongation, contraction, or twisting motions. Using a quantitative model based on lamination theory, we establish design principles for the digital fabrication of silicone-based soft actuators whose functional response is programmed within the material's properties and architecture. Exploring such programmability enables 3D printing of a broad range of soft morphing structures.

[1] Complex Materials, Department of Materials, ETH Zürich, 8093 Zürich, Switzerland. [2] UCD School of Mechanical & Materials Engineering, University College Dublin, Belfield, Dublin 4, Ireland. Correspondence and requests for materials should be addressed to J.A.F. (email: jakob.faber@mat.ethz.ch) or to A.R.S. (email: andre.studart@mat.ethz.ch)

Soft actuators enable smooth and conformable complex motions that ensure safe interactions of robots with humans and have led to impressive assistive technologies for rehabilitation and training[1–4]. Besides conformability and complex motion, low weight and reduced energy consumption are other major advantages of soft actuators compared to conventional rigid counterparts. In contrast to the mechatronic control of conventional actuation systems, soft actuators can be driven by heat, light, air pressure, or liquid displacement. Air- and liquid-driven actuators, also known as fluidic elastomer actuators (FEAs), are particularly interesting because of their simple control without the need of many active components[5]. FEAs inspired by octopus, worms, and starfish have been shown to perform tasks that would not be possible with hard structures[1]. Despite such encouraging developments, further research in materials and fabrication technologies is needed to enable seamless manufacturing of soft actuators that reach the level of motion control observed in biological systems[6].

Using materials and fabrication technologies available since the 1950s, the so-called McKibben's artificial muscle is one of the earliest examples of pneumatic soft actuators that can achieve large contractile and extensional deformations[7]. In this system, an inflatable elastomeric bladder is sheathed with a double helical fiber weave. The strong weave restricts expansion of the bladder in the longitudinal direction, forcing it to expand radially when the system is pressurized. The key design principle underlying such actuation mechanism is the differential deformation of the structure along the longitudinal and transverse directions. Several other combinations of materials and fabrication technologies have explored this principle to build continuum robotic manipulators inspired by muscular hydrostats like tentacles and trunks[6,8–14]. Typically, a soft part comprising pressure chambers is combined with a stiff elastomer with embedded paper, fabrics, or plastic film to achieve asymmetric strain when pressurized[15–17]. Individual parts are often produced separately in sequential molding steps and later assembled to a functional unit[4]. Simple morphologies with one to two materials are preferred to minimize the risk of delamination while still facilitating fabrication and assembly. In addition to the delamination issue, such molding techniques are time consuming, labor-intensive, and limited in terms of overall morphology.

With the advent of two-dimenional (2D) soft lithography and more recently three-dimensional (3D) printing technologies, a wider variety of geometries and materials became accessible for the fabrication of soft actuators[1–5,18,19]. Because of their ease in processing, biocompatibility, and high stretchability, silicones have been the material of choice in this newer generation of actuators[2,4]. In this context, 3D printing has first been used to generate negative molds into which silicones are cast and consolidated[16,20]. More recently, 3D printing of silicone-based materials using stereolithography and Direct Ink Writing (DIW) have been shown[3,4,19] and used for the digital manufacturing of soft actuators inspired by muscular hydrostats[1]. For example, elastomeric materials have been 3D printed using capillary suspension inks containing polydimethylsiloxane (PDMS) in the form of both pre-cured microbeads and uncured liquid precursor dispersed in water[4], or using room-temperature vulcanizing silicones[20] as well as thermoplastic elastomers suitable for Fused Filament Fabrication[21]. However, current 3D printed soft actuators have thus far been limited to bending as the main motion mode and are far from reaching the level of complexity of the programmed architecture found in biological systems.

The fiber architecture of many biological actuators like the elephant trunk, the mammalian tongue, and octopus arms exploits the anisotropic local deformation induced by an external driving stimulus as a means to achieve programmable motion[6,22]. The driving stimulus in such muscular hydrostats is an electrical impulse emitted from a main processing unit, namely the brain. As opposed to recent examples of FEAs, the directionality is not programmed through the design of fluidic channels. Instead, muscular hydrostats achieve directional deformation through the architecture of electro-responsive muscle fibers around the biological actuator. Embedding morphological information inside the material through the local fiber architecture reduces the number of active components necessary for continuous motion, thus saving time and energy associated with signal processing in the brain[6]. This contrasts with the conventional mechatronic control of actuators, which requires more intensive central processing to impart complex continuous motion. Remarkably, biological structures exceed conventional robots in complexity of motions and adaptability by employing solely compliant materials. Using a similar concept, plants exploit fiber architecture to achieve anisotropic deformation and thus grow or change shape in pre-programmed geometries[23–25]. In contrast to muscular hydrostats, the fibers passively restrict the water-driven expansion of a continuous matrix in well-defined directions to achieve differential deformation of the material. Despite these differences in the active driving material, the relative dimensional changes between matrix and fiber are the same, thus leading to an equivalent final motion. Implementing such bioinspired strategies in fluidic elastomer actuators using a 3D printing platform would be an effective approach to digitally fabricate compact soft actuators with complex motion programmed within the material.

In this study, we present a multimaterial 3D printing platform for the digital fabrication of silicone soft actuators displaying a wide range of motions that are programmable within the material's bioinspired fiber architecture. In contrast to previous work, the proposed approach can be utilized to directly print silicones with a wide range of elasticities without requiring separate casting and assembly of silicone elements. Seamless multimaterial 3D printing of silicones with tuneable elasticity massively simplifies the fabrication process of soft robots by obviating the need for assembly, thereby reducing the risk of interfacial delamination while further extending the freedom of form. These capabilities are demonstrated by 3D printing functional pneumatic soft robots with precisely pre-programmed architectures that lead to complex shape transformations.

## Results

**Bioinspiration and silicone 3D printing platform.** The fibrous architecture of the muscular hydrostats that inspired our soft actuators leads to an anisotropic strain upon an electric stimulus (Fig. 1a). In our design, the multimaterial architecture leads to anisotropic strain upon pressurization (Fig. 1b). In both cases, the anisotropic strain is then transformed into complex motion modes. To implement the differential deformation design concept underlying the actuation principle of muscular hydrostats, stiff silicone stripes are printed on top of a soft silicone cylinder. Changing the lead angle of the stiff stripes relative to the longitudinal axis of the cylinder leads to fiber architectures analogous to those of the biological counterpart. The seamless production of four functional pneumatic soft robots with continuous programmed actuation is demonstrated using photocurable silicone inks with tuneable elasticity, called 'Silinks'. The Silink platform enables multimaterial DIW of light-curable silicones with locally tunable stiffness into soft actuators with high freedom of programmable motion.

The use of the Silinks with thiol-functionalized crosslinkers dispersed in a continuous vinyl-terminated polysiloxane phase (Fig. 1c) allows for bonding of deposited inks to already cured

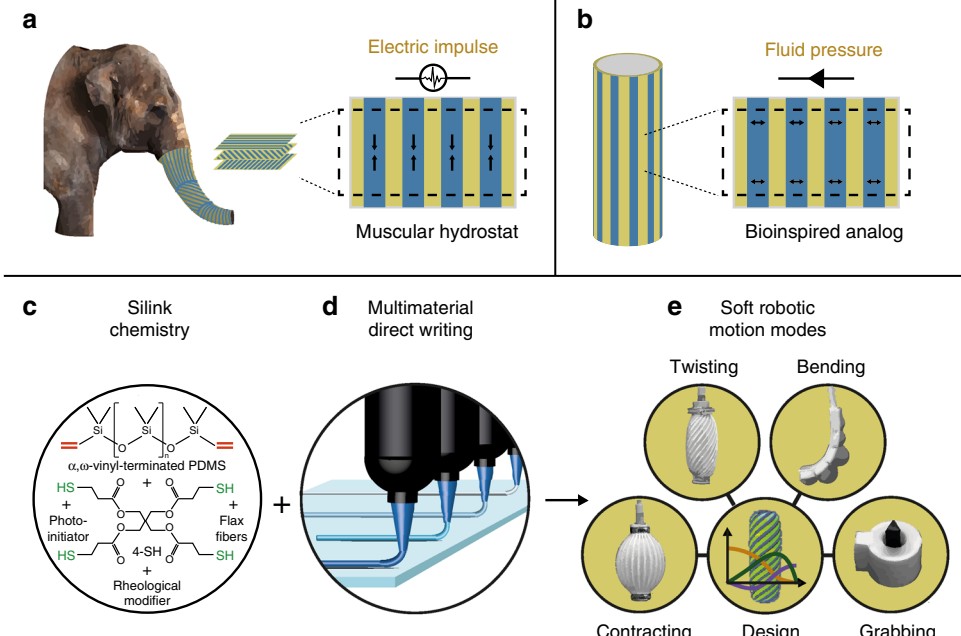

**Fig. 1** Muscular hydrostat and 3D printed bioinspired soft actuators. **a** Multilamellar fiber architecture and functional principle of the elephant trunk used here as an example of a muscular hydrostat. Muscle fiber contraction (blue) driven by neuron impulses causes anisotropic contraction of individual lamellae. The elephant image has been adapted from Digital Zoo/Photodisc/Getty. **b** Bioinspired analog of muscular hydrostats and plant cell walls comprising linear stiff stripes (blue) separated by a soft matrix (yellow). Anisotropic deformation in this case is driven by liquid displacement or air pressure and results from the preferential reinforcement parallel to the stiff stripes. **c** Vinyl-terminated silicones are blended with a multivalent thiol-crosslinker, fumed silica as rheological modifier and a photoinitiator to generate light-curable silicone inks, named Silinks. Variable ink constituents yield Silinks with tunable stiffness ranging from soft and stretchable to hard and stiff. **d** Multimaterial 3D printing seamlessly combines different stiffness in a single print, allowing for precise programming of the actuator's shape transformations when inflated (**e**)

---

### Table 1 Formulations for the soft, intermediate, and stiff Silinks

| Components | Soft ink | Intermediate ink | Stiff ink |
|---|---|---|---|
| Silicone | 1 g Ecoflex 00-30 A | 0.5 g Dragonskin 30 A | 0.5 g Sylgard 184 |
| Vinyl Q-Resin Dispersion VQM-135 | — | 0.5 g | 0.5 g |
| Pentaerythritol tetrakis (3-mercaptopropionate) (4SH) | 0.03 mL | 0.23 mL | 0.5 mL |
| Hydroxycyclohexyl phenyl ketone (HHPK) | 0.03 g | 0.03 g | 0.03 g |
| 9-Vinylcarbazole (enhancer) | 0.005 g | 0.005 g | 0.005 g |
| Wacker HDK Fumed silica H18 | 0.05 g | 0.05 g | 0.1 g |
| Flax fibers | — | — | 0.15 g |

---

sections and thereby leads to perfect interface strength. Not only is this interface strength present between sections that were cured at different times, but also between different Silink formulations. This ideal bonding in multimaterial structures opens the possibility to directly 3D print silicones with different local materials properties and geometries (Fig. 1d). Four designs of pneumatic soft robots including a bender, a grabber, a twister, and a contractor are introduced to illustrate the potential of the Silink platform (Fig. 1e).

**Light-curable silicones with tuneable elasticity.** To obtain photocurable resins that can be printed by multimaterial DIW, viscoelastic silicone inks that form an emulsion and result in elastomers with variable stiffness after polymerization were developed. Tuneable mechanical properties are achieved by blending vinyl-terminated silicones of different molecular weights with a multivalent low molecular weight thiol-crosslinker (4-SH)

at distinct ratios. The vinyl-terminated silicone forms the continuous phase of the resulting emulsion, whereas the thiol crosslinker remains as homogeneously distributed disperse phase. The process is solvent free and does not require chemical modification of the constituents. Three distinct ink designs, hereafter named soft, intermediate, and stiff Silinks, were formulated (Table 1). The Silinks contain the vinyl-terminated silicone, the thiol crosslinker, and fumed silica as rheological modifier. The stiff Silink contains in addition 15 wt% of chemically functionalized short flax fibers with lengths in the range of 300–400 μm.

The inks were first characterized with respect to their viscoelastic properties to determine if they fulfill the rheological requirements for multimaterial DIW. Strain amplitude sweeps obtained from oscillatory rheological measurements indicate the presence of a predominantly elastic network at rest and a storage modulus plateau $G'$ of 2, 5, and 43 kPa for the soft, intermediate, and stiff inks, respectively (Fig. 2a). According to theoretical

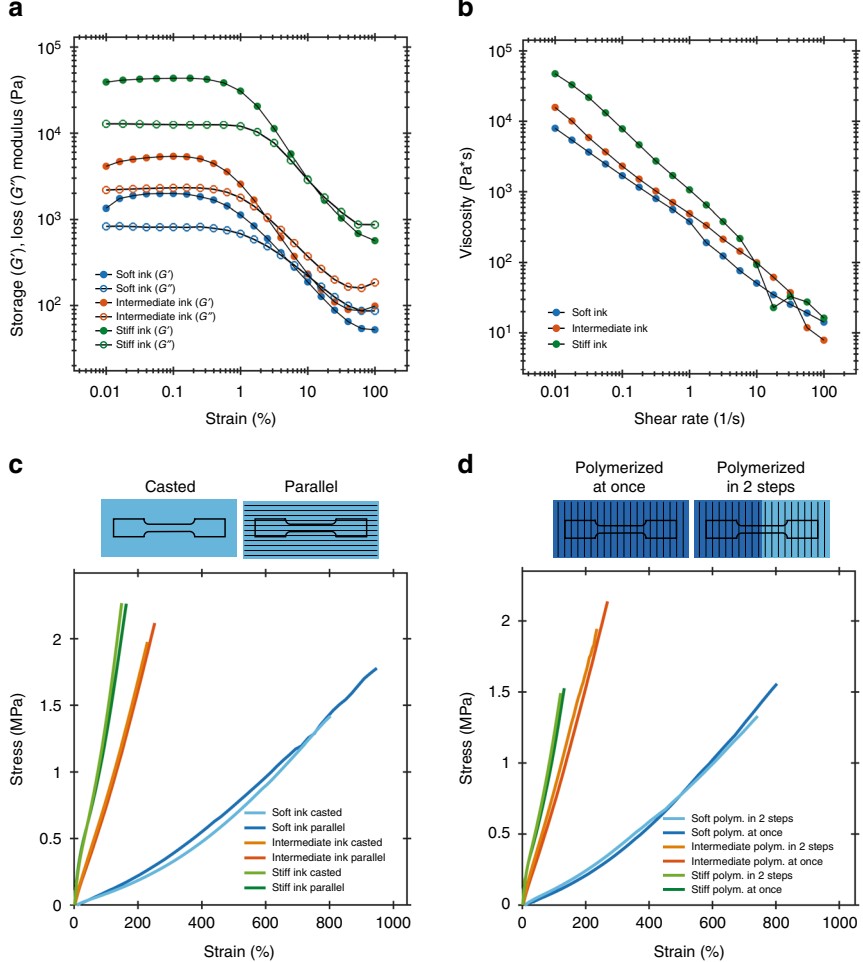

**Fig. 2** Rheological properties of Silinks and mechanical behavior of printed inks after polymerization. **a** Storage modulus, $G'$, and loss modulus, $G''$, of the soft, intermediate, and stiff Silink as a function of the strain amplitude applied in an oscillatory sweep measurement. Crossovers at high strains indicate the presence of a predominantly elastic network at rest, which is crucial for shape retention after extrusion. **b** All Silinks exhibit shear-thinning behavior visible by the decrease in viscosity at higher shear rates under steady-state conditions. In combination with the viscoelastic network formation, shear thinning guarantees a consistent flow through the nozzle and enables high printing accuracy with a spatial resolution as low as 300 μm. **c** Soft, intermediate, and stiff Silinks tested parallel to the printing direction and compared to casted samples. Printing does not adversely affect the rupture strengths of the Silinks. **d** Influence of the polymerization protocol on the bonding strength between printed filaments. Continuous films polymerized post printing are compared to a film with polymerization halfway through the print. The bonding strength between filaments directly polymerized after deposition is not affected in comparison to a film deposited continuously in one run and polymerized after printing. All samples were tested perpendicular to the printing direction

calculations based on simple beam theory[26,27], the measured storage modulus of each Silink formulation should allow for printing of grid structures with free-spanning bridging lengths of 19 (stiff), 13 (intermediate), and 10 mm (soft) using a 0.41-mm nozzle diameter. Crossovers of $G'$ and $G''$ at high strains also indicate the presence of a dynamic yield stress, $\tau_y$, for all the investigated inks. $\tau_y$ values of 24, 35, and 420 Pa were measured for the soft, intermediate, and stiff ink formulations, respectively. Our printing experiments reveal that such yield stress levels are sufficiently high to minimize distortion due to capillary forces and thus ensure instantaneous shape retention with printing accuracy down to 300 μm. Yield stresses above the threshold of approximately 100 kPa required to print overhangs[28] can be achieved by adding higher amounts of fumed silica to the ink formulations. Besides oscillatory measurements, steady-state rheology was also used to characterize the flow properties of the inks. All formulations show a strong shear thinning behavior (Fig. 2b), which leads to a decrease in apparent viscosity by four orders of magnitude for increasing applied shear rates. This facilitates the extrusion process and reduces the pressures

required for printing. Finally, the inks do not show any visible inhomogeneity over weeks when stored under ambient conditions, suggesting no aging or phase separation that could compromise the quality and reproducibility of the formulations.

The mechanical properties of silicone parts printed from different inks were quantified to ensure that the stiffness ratio between the stiff stripes and the matrix (Fig. 1b) is sufficiently high to enable anisotropic deformation of the bioinspired architectures upon pressurization. Indeed, the stiffness changes by one order of magnitude with increasing amounts of crosslinker in the Silinks. The intermediate and stiff formulations reach an elastic modulus of $1.00 \pm 0.06$ and $3.40 \pm 0.17$ MPa, respectively, whereas a value of $0.13 \pm 0.01$ MPa was measured for soft inks (Fig. 2c). Higher stiffness values were accompanied by a decrease in elongation at break ($\varepsilon_B$). $\varepsilon_B$ values of $922 \pm 25\%$, $267 \pm 20\%$, and $150 \pm 11\%$ were measured for the soft, intermediate, and stiff formulations, respectively.

The high extensibility achieved makes the silicone formulations ideal for the fabrication of conformable soft actuators. To evaluate

whether the 3D printing process does not introduce defects that could compromise the mechanical properties of the silicones, the mechanical properties of printed silicone specimens were compared to those of samples directly cast into rectangular molds using the same ink formulations. Printed Silinks show elastic modulus, rupture strength, and elongation at break that are higher or comparable to their casted references when tested parallel to the printing direction (Fig. 2c). This confirms that the printing process does not affect the mechanical properties arising from the Silink chemistry, allowing printed parts to fully benefit from the high stretchability and tuneable stiffness of the silicones used.

To assure soft actuators do not rupture when inflated, a mechanically strong interface between single print lines as well as between different material compositions is also crucial. We observed that all Silinks show strong interfacial bonding between printed filaments. This is demonstrated by the minor deviation in interfacial strength observed when testing a newly printed Silink layer adjacent to an already polymerized layer as compared to a continuous layer that is polymerized at once (Fig. 2d). We assume that the strong interfacial linkage observed for all Silinks originate from chemical bonds between the single lines, regardless of whether they were readily polymerized or not. This suggests that thiol-functional groups are left on the surface after photopolymerization, providing chemical anchors for the next filament to bond to. Only chemical bonding between the extruded filaments enable a layer-by-layer fabrication without risking delamination between the individual layers. Therefore, the described fabrication process is not restricted in its overall size by limited UV penetration of a few millimeters at 365 nm since multiple layers can be sequentially polymerized without minimizing the overall performance.

Despite the strong interfacial bonding between printed filaments, the mechanical properties of the fiber-containing stiff ink were found to depend on the testing orientation relative to the printing direction (Fig. 2c, d). High shear stresses in the nozzle during printing are expected to align the fibers of the stiff ink along the flow direction[6,7,29,30]. Fibers aligned parallel to the tensile loading can take more of the applied mechanical stress, which explains the 14% higher elastic modulus of samples measured in this orientation. By contrast, fibers oriented perpendicular to the applied force introduce critical defects in the material that reduce the rupture strength by 30%. Such anisotropic mechanical properties are strictly connected to the presence of fibers, since the intermediate and soft Silink containing no fibers show equal strengths and stiffness when tested either parallel or perpendicular to the printing direction (Fig. 2c, d).

**3D printing of pneumatic silicone actuators**. Because of their tuneable elasticity combined with strong interfacial bonding, Silinks can be used to 3D print silicone parts with bioinspired fiber architectures for programmable pneumatic actuation. To illustrate these capabilities, cylindrical soft actuators with three distinct fiber architectures were designed and 3D printed using the multimaterial DIW process. In these first three designs, the soft ink formulation is used to generate a highly stretchable cylindrical part, whereas the stiff ink is utilized to create the different programmable fiber architectures on the surface of the soft cylinder. The main design parameter in these examples is the lead angle, $\alpha$, of the stiff stripe relative to the long axis of the underlying cylindrical part. Following the design principle of muscular hydrostats and plant cell walls (Fig. 1a, b), the $\alpha$ values were deliberately chosen to achieve twisting and contraction as exemplary bioinspired actuation modes.

In the printing process, the soft cylinder is first created by depositing a single 0.8 mm thick layer of the soft ink on top of a rotating cylindrical support with an inner diameter of 22 mm. After consolidation by photopolymerization, the soft cylindrical part is then coated with stiff stripes oriented at an angle $\alpha$ relative to the longitudinal axis. Fibers with well-defined lead angles are generated by continuously turning the cylindrical part around its long axis while the stiff ink is deposited through a translating nozzle. Stiff stripes are printed in multiple steps by the additive deposition of 0.4 mm thick layers until a final height of 4 mm is reached. Each layer is directly polymerized in a nitrogen-saturated atmosphere for 1 min after printing.

Pressurization of the bioinspired soft actuators eventually leads to the large-range twisting and contractile motions programmed through the lead angle, $\alpha$. Winding the stripes around the tube at a lead angle of 45° (Fig. 3b) or 30° (Supplementary Note 1, Supplementary Figure 1) resulted in twisting of the tube, in line with the design principle underlying the actuation of muscular hydrostats. The degree of rotation at increasing pressures up to 6 kPa is illustrated below each frame shown in Fig. 3b. Maximal torsions of 225° and 160° were obtained for lead angles of 45° and 30°, respectively. Further reducing the lead angle to 0°, with the stripes now parallel to the main axis, resulted in contraction by radial expansion of the tube when pressurized (Fig. 3a). The contractor displays a maximal contraction of 8.5% at a pressure of 6 kPa and an actuation speed of up to 18 mm/s (Supplementary Figure 4).

**Design principles and modeling of soft actuators**. Understanding the effect of the applied pressure on the large-range motion achieved by the soft actuators is crucial for their design, programming, and successful operation in robotic applications. For the preliminary design of motion modes and a knowledgeable overview of the influencing parameters, simplified analytical models are ideal. Assuming that the stiffness parallel to the fibers is dominated by the stiff stripes and that the soft transverse behavior is dictated only by the high compliance of the soft layer, one can use established lamination theory to obtain the strains underlying all motion modes, namely contraction, elongation, and twisting, as shown in Fig. 4a. For the exemplary twisting mode, this approach results in the following simple relation:

$$\theta = \frac{2L}{D_m} \gamma_{xy} \text{ and } \gamma_{xy} = \frac{p_i D_a}{2t_a} \left( \frac{1}{E^*_{Stiff}} - \frac{1}{E^*_{Soft}} \right) \cdot \left( 3\sin^3(\alpha)\cos(\alpha) + \cos^3(\alpha)\sin(\alpha) \right), \tag{1}$$

where $\alpha$ is the lead angle, $p_i$ is the applied inner pressure, $L$ is the length of the cylinder, $t_a$ and $D_a$ are the average thickness and diameter, and $E^*_{Stiff}$ and $E^*_{Soft}$ are the normalized elastic moduli of the stiff and soft phases, respectively. These relations are derived for all main motion modes by homogenizing the material properties of the fiber architecture and should only be valid for small pressures and deformations (see Supplementary Methods).

With such analytical relationships at hand, a wide design space of parameters can be quickly investigated (Fig. 4a). As expected and confirmed by the experiments, the stripe design parameter $\alpha$ determines the maximum strain achievable, thus defining if the dominant motion mode is contraction (small angles), extension (large angles), or twist (medium angles). The maximum twist per applied pressure occurs not at $\alpha = 45°$, as one might intuitively think, but at 55°, which agrees to previous results obtained with a different route[22].

Insights on the correlation between material properties and final actuation are also possible using the derived analytical equations. To achieve maximum twist per applied pressure, a

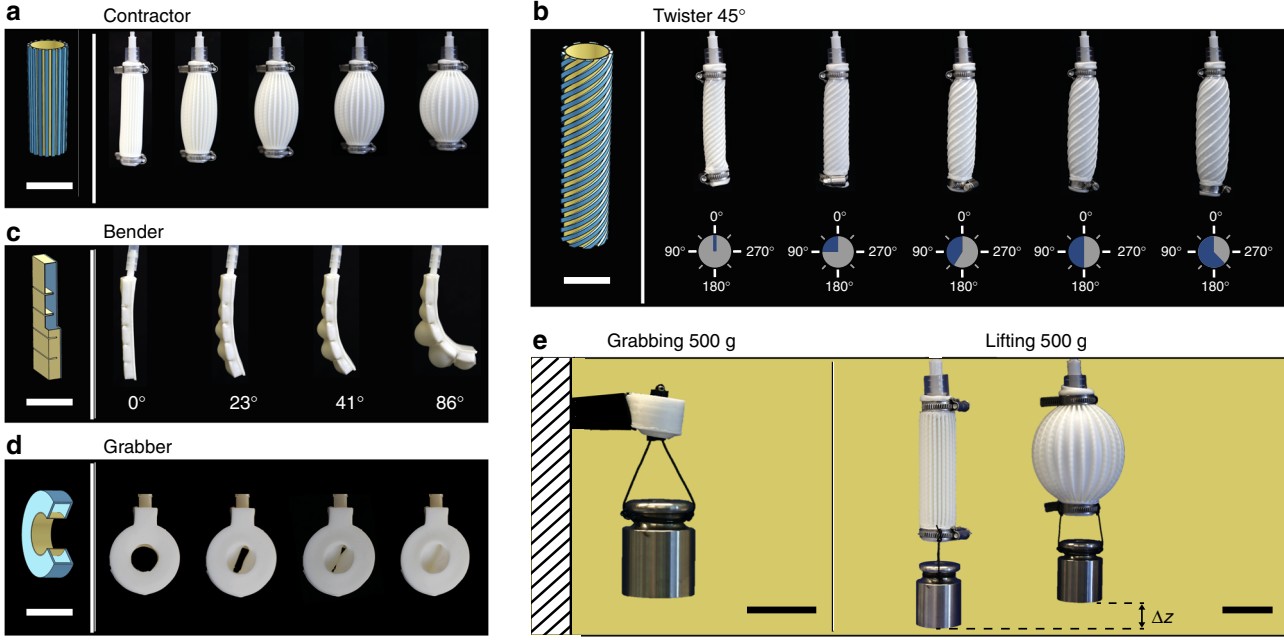

**Fig. 3** Soft actuators with programmed motion modes fabricated by multimaterial 3D printing of silicones. Actuators with bioinspired architectures are shown in **a** and **b**, whereas other multimaterial morphing configurations are displayed in **c** and **d**. **a** Contraction is achieved by printing stiff stripes along the long axis of a soft silicone tube (lead angle $\alpha = 0°$), thereby restricting its elongation and only allowing for radial expansion. **b** A twisting motion is generated by winding the stripes around the soft inner tube with a lead angle of for example 45° with respect to the long axis. The twisting angle increases with the applied internal pressure. **c** Bending motion is obtained when a soft silicone with embedded air cavities is printed onto a stiff silicone film, restricting one side of the chambers from expansion when pressurized. Bending angles up to 90° are achieved at a pressure of 6 kPa. **d** A grabbing and sealing soft actuator was designed by printing a closed, stiff silicone cylinder with a concentric inner soft tube separated by an air cavity. When inflated, the inner tube deforms uniformly until mechanical instability leads to buckling and finally to a complete sealing of the central opening. **e** Load bearing capacity of grabbing and contractile soft actuators. Scale bars are 2 cm for **a**–**d** and 4 cm for **e**

stiffness reduction along the transverse direction is much more efficient than the stiffening of the restricting stripes, as compared to the initial design. This reduction in transverse stiffness can be obtained by choosing for example softer inks or a thinner base cylinder. For the 45° prototype shown in Fig. 3b, our simple linear model predicts a twist of 48° per kPa of applied pressure. The experiments confirm the model relatively well for small pressures and angles (Fig. 4b). This tool is therefore highly valuable in the preliminary design of soft robotic motion and to obtain scaling relations valid for cylindrical geometries and small deformations.

For the large strains and the complex motion patterns made possible by 3D printing of multimaterial actuators, higher-order models are needed for detailed design. Here, we use finite element analysis simulations to model the large nonlinearities of the twisting soft actuator, as illustrative example (Fig. 4c). We chose the twisting mode as an example because of the highest complexity of this motion. The motion of the twister involves anisotropic deformation of the patterned material, large shear strains as well as reorientation of the stripes during twisting. In contrast, grabber and bender are comprised of isotropic regions only and the contractor deforms without shear strain or stripe reorientation, making them the less complex cases. The procedure described here can be easily expanded to these simpler motion modes. Our finite element model of the 45° twister was built according to the measured geometry of the printed prototype. The geometry was virtually inflated by simulating an incompressible fluid inside the cavity. The simulated cavity consists of shell elements for the thin, tubular base layer made from the soft ink. This base layer is attached to solid continuum elements for the reinforcing stripes made from the stiff ink. For each type of ink, a distinct material model was employed to capture the complex

geometrical and material nonlinearities of the different silicone formulations used.

The nonlinear material stress–strain curves strongly influence the stability of inflated large-strain cavities, a problem typical for rubber balloons and their analysis[31]. This link of material properties and balloon stability determines the controllability of the actuators. To achieve controllable motion of a soft robot, a monotonous, constantly rising relationship between applied pressure and resulting displacement is necessary. In inflated cavities, it is known that linear and digressive stress–strain behaviors lead to instabilities during inflation, making the material unsuitable for pressure-controlled motion. In contrast, a monotonous and thus stable relationship between pressure and twist angle occurs for significantly progressive material behavior[32]. The highly progressive stress–strain characteristics of Silinks (Fig. 2) and their structured topology make them especially suitable for these pressure-controlled applications. Indeed, no pressure instabilities were observed in the experiments.

The FEA model provides a good quantitative prediction of the experimental data, particularly in the high-pressure range not covered by the analytical relations (Fig. 4c). The FEA tends to overestimate the stiffness of the system. As a result, the difference between experiments and simulations is maximal at low pressures reaching 35% at 1.2 kPa. This error decreases to 4.5% at the higher pressure of 6 kPa (Fig. 4d). These disagreements are mainly attributed to thickness inhomogeneity arising from the printing process and simplified boundary conditions of the complex, clamped end region of the soft actuators. Improved accuracy might be possible by performing multiaxial material testing, contact simulation of the clamped regions and refinement

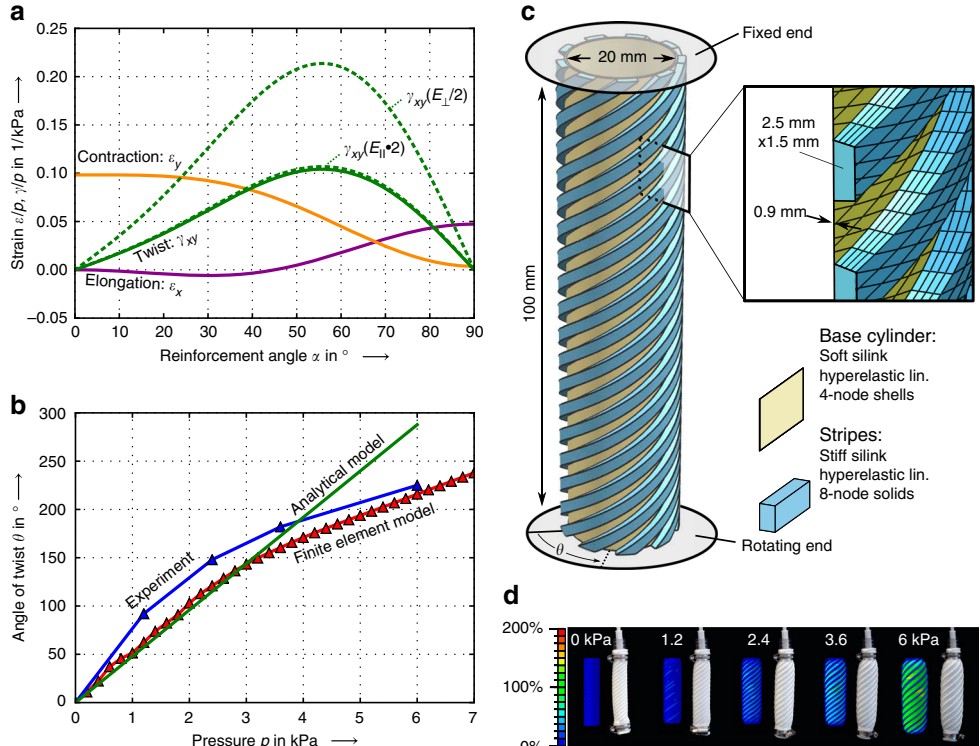

**Fig. 4** Model predictions of the morphing behavior of soft actuators exhibiting bioinspired multimaterial architectures. **a** Analytical prediction of strains over the lead angle of the stiff stripes in a pressurized cylinder. Small angles lead predominantly to contractile motion, large angles to elongation, and diagonal angles to twist. To increase the efficiency of the motion mode twisting, a reduction of the transverse stiffness by 50% is significantly more effective than doubling the parallel stiffness. **b** Predicted and experimentally measured twist angle as a function of the applied internal pressure. **c** Geometry, simulation parameters, and finite element model of the 45° twister. **d** Calculated first principal strain distribution and soft actuator shapes compared to experimental photographs

of the printing process. Smearing of properties into orthotropic homogeneous metamaterials can also reduce the computing power significantly. Overall, the combination of FEA with an accurate material model enables precise prediction of the pneumatic morphing of silicone multimaterial architectures. By pre-programming shape change on a material level and simultaneously using the design freedom of 3D printing, tunable motion modes ranging from basic movements to freeform morphing of great complexity can be achieved.

Importantly, the Silink chemistry proposed here can be utilized for the multimaterial 3D printing of soft actuators beyond the fiber architectures inspired by muscular hydrostats and plant cell walls. We illustrate this by designing soft actuators that exploit silicones with different elastic moduli to program bending and grabbing functionalities. These soft robotic elements were 3D printed by depositing soft and stiff Silinks onto a water-based support ink that was later removed to generate an air cavity. Bending actuators were created by printing the walls of the air cavity using the soft ink except for its bottom part, which was made out of stiff Silink (Fig. 3c). This asymmetry in local elastic modulus resulted in preferential extension of the soft side, allowing the structure to bend towards the stiff side. A maximal bending angle of 90° was achieved in this case at a pressure of 5 kPa. The asymmetric local elastic modulus required for bending can also be programmed by printing stiff stripes with lead angles of 0° and 90° on each side of a soft cylindrical substrate, as illustrated in Supplementary Figure 3. In another example, tuneable geometry and local materials properties were exploited to create actuators with grabbing and sealing functionalities. The actuator consists of a soft inner silicone tube embedded within a concentric outer stiff cylinder (Fig. 3d). To obtain this

multimaterial actuator, two concentric cylinders were printed with an elastomeric inner wall and an outer stiff wall. An air cavity separates the inner from the outer cylinder, which has an inlet channel that connects the actuator to the external pressure system. When inflated, the inner tube first deforms uniformly until mechanical instability leads to buckling and finally to a complete sealing of the central channel. Such a grabbing pneumatic actuator holds and conforms to objects with different masses and shapes, making it a useful tool to handle fragile object or to control mass flow through the central channel (Supplementary Figure 2). Circular, hexagonal, rectangular, and triangular-shaped profiles are grabbed and tightly enclosed by the soft robotic element.

Besides their shape conformability and smooth morphing capabilities, the printed actuators can also carry high mechanical loads. Remarkably, the grabbing and contractile soft actuators are capable to lift a load of 500 g, which corresponds to more than 50 times their body mass. They also operate reliably without losing performance in actuation or show signs of inelastic deformation after extensive lifting cycles. In the experiments shown in Fig. 3e, the load was held in place for 15 min without observing slipping or inelastic deformation of the actuators. Combining the grabber with the contractor, handling of fragile objects with different masses and shapes is possible without losing performance even under heavy loads.

In summary, we developed a 3D printing platform for the digital fabrication of silicone-based soft actuators whose pneumatic-driven motion is programmed within their multi-material architecture. Alike plant systems and muscular hydro-stats, programmable contractile, expanding and twisting motions are achieved by simply controlling the lead angle of the stiff phase

with respect to the long axis of the soft cylindrical actuator beneath. The lead angle determines the local anisotropic strains developed throughout the cylindrical actuator during pressurization, which is the fundamental design principle underlying the programmable motion of these systems. Predicting such local strains with the help of lamination theory and FEA modeling enables mapping of the design space available for programming the different actuation modes. In addition to the lead angle, the relative density of stiff stripes, the elastic modulus ratio between stiff and soft phases, and the aspect ratio of the cylinder are the other relevant design parameters for the soft actuators investigated in this study. Extending these concepts to more complex geometries and multimaterial architectures will enable the digital fabrication of soft actuators with more elaborate programmable motion to fulfill the increasing demand for robots that interact safely with humans.

## Methods

**Materials**. EcoFlex 00–30 A and DragonSkin 30 A were purchased from Smooth-On (USA), whereas Sylgard 184 was acquired from DowCoring (USA). Pentaerythritol tetrakis(3-mercapto-proprionate) (abbreviated as 4SH) and 9-vinylcarbazole (abbreviated as enhancer) were purchased from Sigma-Aldrich (Switzerland). 1-Hydroxycyclohexyl phenyl ketone (abbreviated as HHPK) was provided by TCI (Japan) and silanol-trimethylsilyl modified Q resin (abbreviated as Q resin) was purchased from Gelest Inc. (USA). Hydrophobic fumed silica HDK H18 and hydrophilic fumed silica V15 were supplied by Wacker (Germany) and the short flax fibers were provided by Ruthmann (Germany). For the fiber hydrophobization chlorotis(trimethylsiloxy)silane (abbreviated as CTS) from ABCR (Germany) was used combined with pyridine 95% from Sigma-Aldrich (Switzerland).

**Hydrophobized flax fibers**. Flax short fibers were functionalized using chlorotris(trimethylsiloxy)silane (abbreviated as CTS) in pyridine as a solvent. To functionalize 1 g of flax fibers, 1 mL of CTS and 5 mL pyridine were put into a sealed 50 mL falcon tube with nitrogen-enriched atmosphere and thoroughly mixed. The slurry was shaken for 10 min and dispersed in an ultrasonic bath for 20 min. The fibers were filtered and washed with 60 mL of double distilled $H_2O$. The fibers were dried at 50 °C under reduced pressure and sifted through a 280-µm sieve.

**Ink preparation**. The base component of the silicone rubbers was mixed with 4SH, HHPK (3 wt%), Enhancer (0.5 wt%), and hydrophobic fumed silica to obtain printable inks. Three inks were developed to cover a broad stiffness range: one soft, one intermediate, and one stiff. The soft ink was based on EcoFlex 00–30 A to which 4 wt% of 4SH and 5 wt% hydrophobic fumed silica were added. The intermediate ink contained a mixture of Q resin and DragonSkin 30 A at a mass ratio of 1:1 to which 18.5 wt% 4SH and 5 wt% hydrophobic fumed silica were added. The stiff ink was composed of a mixture of Q resin and Sylgard 184 base at a 1:1 mass ratio to which 33 wt% 4SH and 10 wt% hydrophobic fumed silica were added. For further reinforcement 15 wt% functionalized flax short fibers (280 µm) was also added to the stiff ink. The silicone base, 4SH, HHPK, and Enhancer were mixed with a planetary mixer (ARE-250, Thinky, USA) for 5 min at 2000 rpm, followed by the addition of the hydrophobic fumed silica. The ink was then mixed and degased thoroughly for 5 min at 2000 rpm and 5 min at 2200 rpm, respectively. A support ink containing 2 wt% hydrophilic fumed silica, 2 wt% κ-carrageenan, and 2 wt% hyaluronic acid in double distilled $H_2O$ was used. κ-carrageenan and hyaluronic acid were dissolved in double distilled $H_2O$ at 80 °C under stirring at 2000 rpm. After the gel reached room temperature, hydrophilic fumed silica was added and the gel was mixed at 2000 rpm until it became a homogeneous mixture. The gel was cooled to 5 °C while stirring and as a final step 0.1 wt% hydrogen peroxide was added to prevent bacterial growth.

**Rheology**. To characterize the inks prior to printing, a MCR501 rheometer with cone-plate geometry (CP 25, Anton Paar, Austria) was used. Experiments were conducted at 20 °C. Oscillatory amplitude sweeps at 1 Hz were performed within strain values ranging from 0.01% to 100%, whereas steady-state flow curves were obtained at shear rates varying between 0.01 and 100/s.

**Sample preparation and mechanical testing**. The silicone inks were cast between two glass slides with a thickness of 1 mm or 3D printed using a 3D Discovery (regenHU Ltd., Switzerland). Prior to printing, the inks were filled into cartridges and centrifuged at 3000 rpm for 30 s. Cylindrical needles with an inner diameter of 0.58 mm (H. Sigrist & Partner AG, Switzerland) were used. The extrusion rate was optimized to match a desired printing velocity. To this end, we first printed single lines at constant printing velocities ranging from 4 to 10 mm/s and varied the extrusion rates. The lines were then visually inspected to define optimal printing

speeds between 4 and 10 mm/s and printing pressures in the range 500–4000 kPa. These parameters were adjusted for each of the Silinks. The interface was tested by printing and subsequent polymerization of specimens with a single layer thickness. Following the polymerization, an adjacent second layer was printed next to the already cured layer at the same height. Curing for 60 s was performed using an OmniCure Series 1000 (Excelitas, USA) under nitrogen enriched atmosphere. After the polymerization of the adjacent layer, tensile samples were punched perpendicular to the printing direction. Mechanical tests were performed at 20 mm/min at a pre-force of 0.05 N using an Autograph ASG-X mechanical testing machine (Shimazu, Japan). The elastic modulus was determined between 0.5% and 5% strain. Reported values for ultimate tensile strength, elastic modulus, and elongation at break were averaged over at least five samples.

**3D printing soft robotic actuators**. The bending and grabbing soft robots were printed using the 3D Discovery and cylindrical needles with an inner diameter of 0.41 mm. After each layer, a curing step of 60 s was performed under reduced oxygen atmosphere. After printing, the support was washed with water, leaving air cavities behind. Air tubing was inserted which were then connected to a pressure source. The tubular soft robots for twisting and contraction were printed using a custom built 4-axis CNC gantry printer, which prints on a rotating cylindrical mandrel using an eco-PEN300 (Preeflow, Germany). Cylindrical needles with an inner diameter of 1.19 mm were used and a curing step was performed after each layer. The soft robotics actuators were cured using an OmniCure Series 1000 under reduced oxygen atmosphere for 5 min. Air tubing, mechanical fasteners and polycarbonate profiles were used to make the robots airtight and inflatable.

**FEA simulations**. To account for the sensitivity of the simulations to the shape of the stress–strain curves (see discussion in main text), the Silinks were modeled using a hyperelastic material model as proposed by Marlow[33]. This model was fitted to mechanical characterization results analogous to the ones shown in Fig. 2. To achieve additional accuracy, the soft ink, whose high strains dominate the shape response, was repeatedly pre-stretched to 100% strain before data acquisition to account for the Mullins effect[34]. Furthermore, all strain values were amended using Digital Image Correlation measurements to show the real, highly progressive stress–strain characteristic of the Silinks. As a result, a monotonous pressure–twist relationship was obtained, showing the same digressive behavior as the tested prototype (Fig. 4c, d). All the materials properties and geometrical parameters used in the simulations and in the analytical model are shown in Supplementary Table 1.

**Data availability**. All data are available from the authors upon reasonable request.

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

## Acknowledgements

We thank the financial support from ETH Zürich, the Swiss National Science Foundation (Consolidator Grant number BSCGI0_157696), the European Office of Research and Development (EOARD, grant FA9550-16-1-0007), as well as technical support by the Center for Optical and Electron microscopy of ETH Zürich (ScopeM). This research was also partly supported by the Swiss National Science Foundation through the National Centre of Competence in Research Bio-Inspired Materials.

## Author contributions

All authors designed the research. Experiments were designed by M.S., L.P., and J.A.F. and conducted by M.S. and L.P. The 3D printing setup was developed by F.C. The rheology was measured by P.A.R. The Silink chemistry was developed by M.S. The mechanical models and finite element calculations were developed by J.A.F. The main paper and the Supplementary Information were written by A.R.S., M.S., and J.A.F. All authors discussed the results, their implications, and revised the manuscript at all stages.

## Additional information

**Competing interests:** A patent application comprising the Silinks was filed by M.S., L.P. and A.R.S. The remaining authors declare no competing financial interests.

