## [Peer Review File · Nature Communications]

Reviewers' comments:

Reviewer #1 (Remarks to the Author):

The authors present a bioinspired approach for the fabrication soft robotic actuators based on a novel versatile materials system (which also partly employs flax fibre as a natural resource) together with an advanced 3D printing procedure.

This excellent work is presented and performed on a very high level. The authors apply state of the art methods to investigate the fabricated systems. Additionally, the work is enriched by a quantitative theoretical model for prediction of the actuation behaviour. They could demonstrate that the functional response of their system can be programmed in a wide range of possible movements.

The work provides a general platform and approach for novel bioinspired actuation systems and is therefore recommended for publication.

Reviewer #2 (Remarks to the Author):

The manuscript presents the fabrication of an inflatable pneumatic soft robot, inspired from McKibben's artificial muscle, with the ability of anisotropic actuations manifesting as contractor, bender, grabber, and twister by means of direct ink writing of a custom mixture of materials mainly made of viscoelastic silicon. Photo curing has been used to solidify the deposited resins and a wide range of mechanical properties achieved with different material concentrations. In this work various types of actuations have been demonstrated and finite element analysis of twisting is developed to support the experimental results.

The study has shown effort in developing soft robots with application of additive manufacturing as well as a sound knowledge of material chemistry has been demonstrated. There are however some issues that should be addressed before publication.

➤ The use of the term "3D printing" does not seem to be suitable for this study since I could not see the capability of printing spatial structures with any resolution in terms of porosity or other 3D printing related features. I suggest the use of additive manufacturing instead.

➤ There are other already 3D printed pneumatic actuators particularly utilising silicone materials most of which more or less elicited bending in a programmable way as:

- 3D Printing of a Thin-Wall Soft and Monolithic Gripper Using Fused Filament Fabrication Hisham, M.C.M. Anver, Rahim Mutlu and Gursel Alici.
- Buckling of Elastomeric Beams Enables Actuation of Soft Machines, Dian Yang, Bobak Mosadegh, Alar Ainla, Benjamin Lee, Fatemeh Khashai, Zhigang Suo, Katia Bertoldi, and George M. Whitesides
- Design and Fabrication of a Pneumatic Soft Robotic Gripper for Delicate Surgical Manipulation, Jin Guo¹, Yi Sun, Xinquan Liang, Jin-Huat Low, Yoke-Rung Wong, Vincent Shian-Chao Tay, Chen-Hua Yeow.

What is the novelty of this work compared to them? The distinct feature of this work against recently published work should be highlighted.

➤ In the introduction section, there is part where the authors mention electrical stimulus involved in biological actuation etc. How does this relate to the current work?

➤ In 'Sample Preparation and Mechanical Testing', it is said that the pressure and a range of printing speeds are opted to be in a specific range. How did the authors come up with these values? Were these values determined by trial and error or was there a procedural theoretical mechanism? A systematic approach, considering rheological and mechanical parameters, needed to be described to reach these values.

➤ What was the reason for developing finite element analysis simulations only to model the large nonlinearities of the twisting soft actuator and no other types of actuations presented?

➤ Adding time dependent experimental actuations may be beneficial for giving more insight into the soft actuator behaviour (i.e. end point location vs. time for bending).

Overall the paper is well written and it is a valuable contribution to the emerging technologies of soft actuators and I recommend the acceptance of the manuscript subject the minor editing as noted above.

Reviewer #3 (Remarks to the Author):

This is a hard paper to evaluate. I say this because there exists a Yin and Yang to it that I find difficult to reconcile. The positive aspect is that this is a beautiful work from the perspective of craftsmanship in engineering. The downside, though, is that the advances it demonstrates in core science--whether in terms of mechanical engineering, materials science, or whatever--are actually rather limited. That is the most critical point I have to make--that while the designs that are shown are actually quite beautiful, the mechanics they embed are actually rather obvious/simple.

I believe the paper is best evaluated against its own stated big-picture benchmark. This is stated in the authors own words in the early sections of the papers where they make a generally correct statement that there exists a significant need and opportunity for new methods of fabrication--to realize prospects for construction that are not "hindered by the lack of simple manufacturing routes to generate parts that combine multiple materials with intricate shapes and architectures". The paper really falls short here in a big way. There is really no major advance made in materials chemistry. There is really no big breakthrough in printing. It is, from my perspective, a weak homage to things done by others (notably Jennifer Lewis) that are far more sophisticated. The fundamental features of the materials and printing aspects reported here are thus a bit imitative is how I see it. It is not a persuasive argument for publication here.

Response to referees:

Reviewer #1:

The authors present a bioinspired approach for the fabrication soft robotic actuators based on a novel versatile materials system (which also partly employs flax fibres as a natural resource) together with an advanced 3D printing procedure.

This excellent work is presented and performed on a very high level. The authors apply state of the art methods to investigate the fabricated systems. Additionally, the work is enriched by a quantitative theoretical model for prediction of the actuation behaviour. They could demonstrate that the functional response of their system can programmed in a wide range of possible movements.

The work provides a general platform and approach for novel bioinspired actuation systems and is therefore recommended for publication.

Thank you very much for your positive assessment of our work and the recommendation for publication.

Reviewer #2:

The manuscript presents the fabrication of an inflatable pneumatic soft robot, inspired from McKibben's artificial muscle, with the ability of anisotropic actuations manifesting as contractor, bender, grabber, and twister by means of direct ink writing of a custom mixture of materials mainly made of viscoelastic silicon. Photo curing has been used to solidify the deposited resins and a wide range of mechanical properties achieved with different material concentrations. In this work various types of actuations have been demonstrated and finite element analysis of twisting is developed to support the experimental results.

The study has shown effort in developing soft robots with application of additive manufacturing as well as a sound knowledge of material chemistry has been demonstrated. There are however some issues that should be addressed before publication.

1. The use of the term "3D printing" does not seem to be suitable for this study since I could not see the capability of printing spatial structures with any resolution in terms of porosity or other 3D printing related features. I suggest the use of additive manufacturing instead.

Many thanks for this detailed feedback and your overall positive evaluation of our manuscript. Additive manufacturing and 3D printing have been used as interchangeable terminology in literature to describe the continuous deposition of single layers on top of one another. Nevertheless, additive manufacturing might imply the addition of features onto an existing object. Strictly speaking, the terminology "additive manufacturing" is more precise regarding the presented examples of the twister and the contractor where layers are deposited onto a rotating cylinder. However, the rationale of using the terminology "3D printing" for the bender and the grabber can be explained with the printing of spatial structures in different materials to yield open cavities as well as complex geometries and actuation behavior.

[Redacted]

[Redacted]

2. There are other already 3D printed pneumatic actuators particularly utilizing silicone materials most of which more or less elicited bending in a programmable way as:

- 3D Printing of a Thin-Wall Soft and Monolithic Gripper Using Fused Filament Fabrication Hisham, M.C.M. Anver, Rahim Mutlu and Gursel Alici.
- Buckling of Elastomeric Beams Enables Actuation of Soft Machines, Dian Yang, Bobak Mosadegh, Alar Ainla , Benjamin Lee , Fatemeh Khashai , Zhigang Suo , Katia Bertoldi , and George M. Whitesides
- Design and Fabrication of a Pneumatic Soft Robotic Gripper for Delicate Surgical Manipulation, Jin Guo¹, Yi Sun, Xinquan Liang, Jin-Huat Low, Yoke-Rung Wong, Vincent Shian-Chao Tay, Chen-Hua Yeow.

What is the novelty of this work compared to them? The distinct feature of this work against recently published work should be highlighted.

Thank you very much for the suggested literature. We have now cited these papers in the revised manuscript. The paper by Guo *et al.* uses conventional casting techniques where the mold is 3D printed in an acrylic resin and the silicone is poured in at a later stage. Likewise, Yang *et al.* fabricated the entire structure in several pieces, using silicone elastomer cured in a mold fabricated from acrylonitrile butadiene styrene (ABS) and generated using a FDM 3D printer. In contrast to these conventional silicone castings in 3D printed molds, the presented Silink platform entails the capability of direct printing of light-curable silicones with variable stiffness; thereby it does not requiring casting and assembly of silicone elements.

To clarify and differentiate our work from the mentioned literature, we added and modified the

following paragraph:

Elastomeric materials have been also 3D printed using capillary suspension inks containing polydimethylsiloxane (PDMS) in the form of both pre-cured microbeads and uncured liquid precursor dispersed in water (1), or using Room-Temperature Vulcanizing silicones (2) as well as thermoplastic elastomers suitable for Fused Filament Fabrication (3).

And:

In contrast to previous work, this approach can be utilized to directly print silicones with a wide range of stiffness levels without requiring separate casting and assembly of silicone elements. Multimaterial 3D printing of silicones with tuneable elasticity massively simplifies the fabrication process of soft robots by obviating the need for assembly, thereby reducing the risk of interfacial delamination while even further extending the freedom of form. These capabilities are demonstrated here by 3D printing functional pneumatic soft robots with precisely preprogrammed architectures that lead to complex motion capabilities. Moreover, the use of such Silinks with thiol-functional crosslinkers dispersed in a continuous vinyl-terminated polysiloxane phase allows for bonding of deposited inks to already cured sections and thereby leads to perfect interface strength. Not only is this interface strength present between sections that were cured at different times, but also between different material formulations. This ideal bonding in multimaterial structures opens the possibility to directly 3D print silicones with different local materials properties and geometries. Four designs of pneumatic soft robots including a bender, a grabber, a twister and a contractor are introduced to illustrate the potential of the Silink platform.

3. In the introduction section, there is part where the authors mention electrical stimulus involved in biological actuation etc. How does this relate to the current work?

Thank you for that remark. Our intention here is to point out the analogy between our bioinspired actuators and the muscular hydrostat. Therefore we added the following paragraph to the manuscript:

In our bioinspired designs, the fibrous architecture leads to anisotropic strain upon pressurization, which is analogous to the anisotropic strains generated by electric stimulus in hydrostats. In both cases, the resulting differential deformations are then transformed into complex motion modes.

4. In 'Sample Preparation and Mechanical Testing', it is said that the pressure and a range of printing speeds are opted to be in a specific range. How did the authors come up with these

values? Were these values determined by trial and error or was there a procedural theoretical mechanism? A systematic approach, considering rheological and mechanical parameters, needed to be described to reach these values.

Thank you for these remarks. In a first set of experimental testing, the shear rheological properties of the Silinks were tested under oscillatory (Figure 2A) and steady-shear conditions (Figure 2B). Viscoelastic and shear thinning behavior is required for the ink to flow through the nozzle under shear and a yield stress above 100 Pa*s is needed to retain shape after extrusion as reported in our previous work (4–6). Hence, The rheological properties were tuned with fumed silica to enhance the viscoelasticity of the ink composition necessary for 3D printing. Shear thinning facilitates the flow through the printing nozzle whereas the inks viscoelasticity is crucial for shape. For stacked 3D structures, a yield stress is necessary to hold it's own weight and to prevent shape distortion.

In a next set of experiments, the extrusion rate was optimized to match a desired printing velocity by printing single lines at a constant printing velocity and variable extrusion rate. The lines were then visually inspected to define optimal printing parameters. To clarify the procedure to reach the opted printing parameters, we added the following paragraph to the paper:

The extrusion rate was optimized to match a desired printing velocity. To this end, we first printed single lines at constant printing velocities ranging from 4 to 10 mm/s and varied the extrusion rates. The lines were then visually inspected to define optimal printing speeds between 4 and 10 mm/s and printing pressures in the range 500 - 4000 kPa. These parameters were adjusted for each of the Silinks.

5. What was the reason for developing finite element analysis simulations only to model the large nonlinearities of the twisting soft actuator and no other types of actuations presented?

Thank you for the question. More than showing accordance of all motion modes with their respective simulations, we decided to first show in a simpler analytical model the strains underlying all motion modes, namely contraction, elongation, and twisting, as shown in Figure 4A. With such analytical models, one can identify the parameters that most strongly influence the motion of the actuator. As an example of the insights we obtain from the general analytical model we highlight the following extract from the manuscript:

“Insights on the correlation between material properties and final actuation are also possible using the derived analytical equations. To achieve maximum twist per applied pressure, a stiffness reduction along the transverse direction is much more efficient than the stiffening of the

restricting stripes, as compared to the initial design.”

From this broad coverage of all motion modes using an analytical model, we then funnel down one example in-depth. The intention here is to show the basics widely, and focus on one motion mode to exploit the entire modeling process. This process can be expanded to other motion modes. The reason we chose the twisting mode is because of the highest complexity: only the twister involves the anisotropic patterned material, large shear strains as well as a reorientation of the stripes during twisting. In contrast, grabber and bender are comprised of isotropic regions only and the contractor deforms without shear strain or stripe reorientation, making these modes the less complex cases.

6. Adding time dependent experimental actuations may be beneficial for giving more insight into the soft actuator behaviour (i.e. end point location vs. time for bending).

Thank you for that great input. It is indeed of great value to compare our pneumatic actuators to the state of the art. We added the following paragraph to the supplementary information where we show experimental data on the dynamic actuation behavior of the pneumatic contractor (Figure S4).

Experimental data on the dynamic actuation behavior of the pneumatic contractor (Figure 3A). Using a computer controlled valve, we pressurized the contractor with 0.4 bar at a frequency of 1 Hz. We recorded the contractile displacement at the free end using a laser distance measuring system (Keyence, LK-G5000). The slight deviations in the waveform of individual cycles are attributed to the dynamics of the pressure control system. The observed actuation speeds were 7.5 mm/s during inflation and 18 mm/s during deflation. In both cases, the air supply and hose diameter were the limiting factors. This places the Silink soft robots among the fastest soft robotic systems (7), as there are no intrinsic limits for the actuation speed such as creep, capillary dynamics, or mass and heat transfer.

a)

b)

c)

Figure S4: Dynamic behavior of contracting motion mode. a) Cyclic behavior at actuation using 0.4 bar and 1 Hz. b) Inflation phase at 7.5 mm/s. c) Deflation phase at 18 mm/s.

Reviewer #3:

This is a hard paper to evaluate. I say this because there exists a Yin and Yang to it that I find difficult to reconcile. The positive aspect is that this is a beautiful work from the perspective of craftsmanship in engineering. The downside, though, is that the advances it demonstrates in core science--whether in terms of mechanical engineering, materials science, or whatever--are actually rather limited. That is the most critical point I have to make--that while the designs that are shown are actually quite beautiful, the mechanics they embed are actually rather obvious/simple.

I believe the paper is best evaluated against its own stated big-picture benchmark. This is stated in the authors own words in the early sections of the papers where they make a generally correct statement that there exists a significant need and opportunity for new methods of fabrication--to realize prospects for construction that are not "hindered by the lack of simple manufacturing routes to generate parts that combine multiple materials with intricate shapes and architectures". The paper really falls short here in a big way. There is really no major advance made in materials chemistry. There is really no big breakthrough in printing. It is, from my perspective, a weak homage to things done by others (notably Jennifer Lewis) that are far more sophisticated. The fundamental features of the materials and printing aspects reported here are thus a bit imitative is how I see it. It is not a persuasive argument for publication here.

Thank you for critically assessing our manuscript and for qualifying our paper as “beautiful work from the perspective of craftsmanship in engineering”. We believe a major point has possibly been overlooked here. We refer to the main advancement reported in terms of materials chemistry. The Silink chemistry that we developed allows for the first time the direct printing of silicones with stiffness that is tunable over a broad range without the use of molds or support gels. This has been achieved by developing photocurable inks that can be rheologically modified to be 3D printable, while simultaneously incorporating thiol chemistry. The chemistry is designed such that the thiol groups allow for bonding of deposited ink to already cured sections of the part and thereby leads to perfect interface strength. Not only is this interface strength present between sections that were cured at different times, but also between the different material formulations, leading for the first time to multimaterial printing of silicones. This ideal bonding in multimaterial structures is crucial prerequisite for the operation and longevity of the designs we then created. We deliberately chose the simple cylindrical geometry as a model system and fundamental unit that can be studied in details using analytical modelling. This model system can be expanded to more complex structures that go far beyond these simple geometries. [Redacted]

[Redacted]

Finally, it is important to emphasize that this example can only be manufactured due to our novel Silink chemistry that allows direct multimaterial 3D printing of silicones. To clarify this core claim, we largely modified the respective paragraph in the manuscript, as follows (see also Reviewer 2, Question 2):

In contrast to previous work, this approach can be utilized to directly print silicones with a wide range of stiffness levels without requiring separate casting and assembly of silicone elements. Multimaterial 3D printing of silicones with tuneable elasticity massively simplifies the fabrication process of soft robots by obviating the need for assembly, thereby reducing the risk of interfacial delamination while even further extending the freedom of form. These capabilities are demonstrated here by 3D printing functional pneumatic soft robots with precisely preprogrammed architectures that lead to complex motion capabilities. Moreover, the use of such Silinks with thiol-functional crosslinkers dispersed in a continuous vinyl-terminated polysiloxane phase allows for bonding of deposited inks to already cured sections and thereby leads to perfect interface strength. Not only is this interface strength present between sections that were cured at different times, but also between different material formulations. This ideal bonding in multimaterial structures opens the possibility to directly 3D print silicones with different local materials properties and geometries. Four designs of pneumatic soft robots including a bender, a grabber, a twister and a contractor are introduced to illustrate the potential of the Silink platform.

References:

1. O'Bryan, C. S. *et al.* Self-assembled micro-organogels for 3D printing silicone structures. *Sci. Adv.* **3**, (2017).
2. Yang, D. *et al.* Buckling of Elastomeric Beams Enables Actuation of Soft Machines. *Adv. Mater.* **27**, 6323–6327 (2015).
3. Anver, H. M. C. M., Mutlu, R. & Alici, G. 3D printing of a thin-wall soft and monolithic gripper using fused filament fabrication. in *2017 IEEE International Conference on Advanced Intelligent Mechatronics (AIM)* 442–447 (IEEE, 2017).
4. Kokkinis, D., Schaffner, M. & Studart, A. R. Multimaterial magnetically assisted 3D printing of composite materials. *Nat. Commun.* **6**, (2015).
5. Minas, C., Carnelli, D., Tervoort, E. & Studart, A. R. 3D Printing of Emulsions and Foams into Hierarchical Porous Ceramics. *Adv. Mater.* **28**, 9993–9999 (2016).
6. Sommer, M. R., Schaffner, M., Carnelli, D. & Studart, A. R. 3D Printing of Hierarchical Silk Fibroin Structures. *ACS Appl. Mater. Interfaces* **8**, 34677–34685 (2016).
7. Mosadegh, B. *et al.* Pneumatic Networks for Soft Robotics that Actuate Rapidly. *Adv. Funct. Mater.* **24**, 2163–2170 (2014).

REVIEWERS' COMMENTS:

Reviewer #2 (Remarks to the Author):

The authors addressed all my earlier comments and I am satisfied with the response. However, I have two suggestions:

1. Regarding the justification of using twisting soft actuators for FEA of nonlinearities in comment 5, I suggest incorporation of your recent response into the manuscript.
2. Regarding the Figure S4 in comment 6, it would be better if graphs b) and c) are shown in one figure.

I recommend the paper to be accepted subject to minor editing as suggested above.

Reviewer #3 (Remarks to the Author):

I have to confess that I have been brought around in my assessment based on the authors' responses. There is in fact an important materials chemistry aspect that I think may have been obscured in the contextual focus adopted in this work. The new capacity for net fabrication of elastomeric structures with broadly adjustable attributes of mechanics is both impressive and important. I am persuaded and believe the paper should be accepted for publication.

Reviewer #2 (Remarks to the Author):

The authors addressed all my earlier comments and I am satisfied with the response. However, I have two suggestions:

1. Regarding the justification of using twisting soft actuators for FEA of nonlinearities in comment 5, I suggest incorporation of your recent response into the manuscript.

Thank you for your positive feedback and the time spent with our manuscript. We agree that this will improve the quality of the manuscript and have added the recommended passage into the main text.

2. Regarding the Figure S4 in comment 6, it would be better if graphs b) and c) are shown in one figure.

We agree and altered the figure according to this suggestion.

I recommend the paper to be accepted subject to minor editing as suggested above.

Reviewer #3 (Remarks to the Author):

I have to confess that I have been brought around in my assessment based on the authors' responses. There is in fact an important materials chemistry aspect that I think may have been obscured in the contextual focus adopted in this work. The new capacity for net fabrication of elastomeric structures with broadly adjustable attributes of mechanics is both impressive and important. I am persuaded and believe the paper should be accepted for publication.

Thank you very much for your positive feedback and the time spent with our manuscript. We highly appreciate this open and honest review process, which has helped a lot to improve, clarify and sharpen the scope and claims of the paper.